# Determination of the Optimal In-Feed Amino Acid Ratio for Japanese Quail Breeders Based on Utilization Efficiency

**DOI:** 10.3390/ani12212953

**Published:** 2022-10-27

**Authors:** Lizia C. Carvalho, Tatyany S. A. Mani, Michele B. Lima, Jaqueline A. Pavanini, Rita B. Vieira, Lizandra Amoroso, Edney P. Silva

**Affiliations:** Department of Animal Sciences, UNESP–Universidade Estadual Paulista, Via de Acesso Professor Paulo Donato Castellane, S/N, Jaboticabal 14883-900, SP, Brazil

**Keywords:** dilution technique, exponential model, Goettingen approach, lysine requirement

## Abstract

**Simple Summary:**

Breeder reproductive responses are optimized if nutritional, environmental, and health requirements are adequately met. Thus, the ideal concentration of amino acids in the diet must be obtained to prevent excess or deficiency to the animal. This may occur due to the inefficiency in the production or excessive excretion of nitrogen. Therefore, it is necessary to determine the optimal relationship for this nutrient category. These results contribute to ensuring optimal ratios of essential amino acids in the diets of Japanese quail breeders based on amino acid efficiency.

**Abstract:**

The description of the genetic potential is the first step to estimating amino acid requirements and the ideal amino acid relation (IAAR). The aim of this study was to estimate the parameters that describe the daily maximum theoretical nitrogen retention (NR_max_T, mg/BWkg^0.67^), daily nitrogen maintenance requirement (NMR, mg/BWkg^0.67^), protein quality (*b*), dietary efficiency of the limiting amino acid (*bc*^−1^) and determine the lysine requirement and the IAAR for Japanese quail breeders. Two nitrogen balance assays were performed, one assay using 49 quails distributed in seven treatments (protein levels between 70.1 and 350.3 g/kg) and seven replicates and other assay to determine the IAAR by the use of *bc*^−1^, 12 treatments and 10 replicate, with a control diet (CD) and 11 treatments that had limited essential amino acids by providing only 60% of the CD. The values obtained for NR_max_T, NMR, *b* and *bc*^−1^ were 3386.61, 0.000486 and 0.000101, respectively. The daily intake of Lys was 291 mg/bird day. Lys was set at 100% for determining the IAAR: 87, 67, 21, 117, 96, 66, 142, 39, and 133 for Met + Cys, Thr, Trp, Arg, Val, Ile, Leu, His, and Phr + Tyr, respectively, for Japanese quail breeders.

## 1. Introduction

Laying birds need amino acids for maintenance, tissue growth, and fertile egg production [1]. Deficiencies or excess of nutrients in hens can cause deficiency and toxicity in offspring, respectively [2,3]. For breeders, the evaluation of maximum productive performance responses is also highlighted, which can only be achieved if the nutritional and environmental requirements are met with precision. Therefore, commercial feeds for breeder quail are formulated in accordance with the recommendations for commercial quails [4,5], which results in excessive feeding of amino acids to these breeder quail. Breeder quail can show changes in production, egg weight, and egg mass when compared to commercial quail, which drives the need for further research in the amino acid requirements for these birds [6].

The ideal relationship between amino acids is expressed in relation to lysine (Lys) [7]. In addition, Lys is considered the second limiting amino acid in diets based on corn and soybean meal for poultry [8]. In laying birds, Lys is used for egg formation, and its requirements change according to the number of eggs produced [9]. Lower levels (0.59%) of Lys are responsible for a decrease in the number of ovarian follicles and immature reproductive organs in broiler breeders.

According to Emmans and Fisher [10], determining the description of the genetic potential of an animal is the first step in estimating amino acid requirements. Some researchers [11,12,13,14] have used the “Goettingen approach”, which uses the nitrogen balance assay and Mitscherlich function to interpret the genetic potential for protein deposition, as well as to calculate the required amino acid intake to support protein growth. The method of individual deletion of the test amino acid in the nitrogen balance has been used to determine the ideal amino acid ratio (IAAR) [11,12,14,15]. Samadi and Liebert [11] used the observed slope of the responses between diets to determine the ideal proportions of amino acids. Therefore, this study aimed to estimate the parameters that describe the maximum theoretical daily nitrogen retention (NR_max_T) and the daily nitrogen maintenance requirement (NMR) using the Goettingen model. These were applied to calculate the protein quality (*b*) and dietary efficiency of the limiting amino acid (*bc*^−1^), to determine the daily requirement of lysine (Lys), and to establish the ideal profile of essential amino acids (Lys, Met + Cys, Thr, Trp, Arg, Gly + Ser, Val, Ile, Leu, His, and Phe + Tyr) for Japanese quail breeds, based on the efficiency of dietary protein utilization using the deletion method.

## 2. Materials and Methods

### 2.1. Location and Ethics Approval

Two nitrogen balance trials were conducted in the Poultry Sector of the Animal Science Department of the Universidade Estadual Paulista (UNESP/FCAV) in accordance with ethical standards and approved by the Ethics Committee for the Use of Animals under protocol 012203/17. In assay 1 was the determination of NMR, NR_max_T, and Lys requirement using utilization efficiency, and in assay 2 was the determination of amino acid efficiency of utilization, requirements, and optimal amino acid ratio.

Assay 1

### 2.2. Housing, Animals and Experimental Design

A total of 49 Japanese quail breeders at 14 weeks of age, during the peak laying period, were used. The birds were standardised by weight and egg production and distributed by experimental units. A completely randomised design was used with seven treatments and seven replicates with a bird in each experimental unit. Experiments were conducted in a climatic chamber composed of air conditioning and exhausters that maintained the temperature at 24 °C. The birds were housed in galvanized wire cages measuring 0.26 m × 0.37 m × 0.36 m, equipped with a linear feeder and nipple drinkers throughout the experimental period. The light program maintained throughout the experimental period consisted of 16 h of light and eight hours of darkness.

### 2.3. Experimental Diets

Initially, two feeds were formulated: a formulation with a high crude protein content (HPD) and a relative deficiency in Lys compared to the other amino acids, and a second formulation that was free of protein and amino acids (NFD) were prepared (Table 1 and Table 2). The intermediate experimental levels of Lys were obtained by diluting the HPD with NFD.

The HPD and NFD diets were named N7 and N0, respectively. The N0 and N7 diets were formulated to contain 0% and 1.68% of Lys, respectively, and were diluted in adequate proportions to obtain the increasing levels of Lys, meeting the recommendations of the other nutrients following the methodology described by Rostagno et al. [16]. The treatments consisted of seven increasing levels of Lys: N1: 3.4 g/kg; N2: 5.0 g/kg, N3: 6.7 g/kg, N3: 8.4 g/kg, N5: 11.8 g/kg, N6: 13.4 g/kg and, N7: 16.8 g/kg.

### 2.4. Measurements and Variables Analysed

The experiment occurred over 22 days, with 7 days of adaptation and 15 days of data collection. The maximum consumption per kilogram of metabolic weight (BW^0.67^) was determined during the adaptation period, when the birds were fed ad libitum. In the subsequent period, the diets were supplied based on the kg of BW^0.67^ of each bird. The supply of feed was adjusted after each weekly weighing of the birds. To avoid waste, the feed for each experimental unit was divided into two meals per day. Subsequently, the supply was controlled in the following days of adaptation and collection period. To obtain daily feed intake, leftovers were weighed daily. Birds were weighed at the beginning and end of the experimental period to measure body weight.

Egg production and weight was measured daily to quantify the daily egg output throughout the experimental period. Eggs were collected, identified and frozen at −20 °C. The total excreta were collected in trays adapted under the cages, twice a day, during the 15 days, placed in plastic bags and stored in a freezer (−20 °C) until the end of the period.

### 2.5. Chemical Analyses

The eggs and excreta were homogenized, and a sample was taken for drying in a forced air ventilation oven at 55 °C for 72 h. Subsequently, the eggs and excreta were weighed, and ground in a Thomas-Wiley mill with a 1 mm sieve. Samples of the collected material were used to determine dry matter and nitrogen content by the Kjeldahl method, according to AOAC [17].

### 2.6. Statistical Analysis

The variables analysed were nitrogen intake (NI), nitrogen excretion (NEX), nitrogen in egg output (NMO) and nitrogen deposition (ND). The nitrogen retention (NR, NR = ND + NMO + NMR) represents the total nitrogen retained by the bird. NMR is the minimum daily nitrogen value for maintenance and was obtained by the relation between NEX and NI adjusted by the exponential function:NEX = NMRe*^b^*^×NI^(1)
where *b* is the slope of the exponential function, and e is the Euler’s number (ln). The daily theoretical maximum nitrogen retention (NR_max_T, mg/BW_kg_^0.67^) was estimated by the relationship between NI and ND adjusted by the exponential function:NR = NR_max_T(1 − e^−*b*×NI^)(2)
where NR is the daily nitrogen retention (mg/BW_kg_^0.67^), and *b* is the slope of the nitrogen retention curve.

With the transformation of Equation (2), the calculation of the parameter *b* is performed, determining the quality of the protein:*b* = ln[NR_max_T − ln(NR_max_T − NR)]/NI(3)

The Lys requirement was calculated using the equation:LAAI = (lnNR_max_T − ln(NR_max_T − NR))/(16 × *bc*^−1^)(4)
where LAAI is the daily intake of the limiting amino acid (Lys) (mg/BW_kg_^0.67^), *c* is the concentration of the first limiting amino acid in the protein (g/16 g nitrogen), and *bc*^−1^ is the linear relationship between *b* and *c*, which expresses the efficiency of the amino acid studied (Lys).

The assumptions of homoscedasticity and residual normality were tested. Subsequently, the data were fitted to exponential models using PROC NLMIXED using SAS software (SAS Institute Inc., Cary, NC, USA, 2014, version 9.4), considering a significance of 0.05%.

Assay 2

### 2.7. Housing, Animals and Experimental Design

Experiments were conducted in a climatic chamber composed of air conditioning and exhausters that maintained the temperature at 24 °C. The birds were housed in galvanized wire cages measuring 0.26 m × 0.37 m × 0.36 m, equipped with a linear feeder and nipple drinkers throughout the experimental period. The light program maintained throughout the experimental period consisted of 16 h of light and eight hours of darkness. A total of 120 Japanese quail breeders at 16 weeks of age, during the peak laying period, were used. The birds were standardised by weight and egg production and distributed by experimental units. A completely randomised design was used with 12 treatments and 10 replicates.

### 2.8. Experimental Diets

In this study, a control diet (CD) diet was formulated with all the nutritional requirements for Japanese quail as estimated by Rostagno et al. [16] for commercial Japanese quail because it does not provide nutritional requirements for breeders. Nitrogen and essential amino acids were provided by corn, soybean meal, corn gluten meal, and crystalline amino acids (Table 3).

The other experimental diets, of a total of 11 diets with different limiting amino acids, were obtained by diluting CD using corn starch (Table 4 and Table 5). The dilution was 40% of the amino acid requirement to be evaluated in each treatment, and the other nutrients and energy were recomposed to meet the same CD level, except for the test amino acid, which was depleted by 40%, according to Dorigam et al. [14].

### 2.9. Data Collection

The trial lasted 20 days, with feed supply and data collection, the first seven days being adaptation and 13 days total collection of excreta and eggs. To obtain daily feed intake, feed leftovers were weighed daily. Birds were weighed at the beginning and end of the experimental period to measure body weight. Egg production was measured daily, as well as egg weight, to obtain daily egg output (egg production × egg weight) throughout the experimental period.

### 2.10. Chemical Analyses

The methodology used was the same as assay 1.

### 2.11. Statistical Analysis

The variables analysed were NI, NEX, NMO and ND, with subsequent calculation of NR. Data were subjected to homoscedasticity of variance and error normality tests. Upon satisfying the premises, analysis of variance, declared as significant at 0.05, was performed, and when treatment effects were detected, Dunnett’s test was applied for all variables. All data were analysed using SAS software (v.9.4; SAS Institute Inc., 2014).

The ideal proportion of amino acids (IAAR) was determined from the limiting diets in which the tested amino acid was reduced by 40%. The determination of the IAAR by the slope of the linear function (*bc*^−1^), expressed as feed efficiency of the limiting amino acid under study, where protein quality (*b*) in each treatment was obtained by Equation (3) by Samidi e Liebert [13]:*b* = (lnNR_max_T − ln (NR_max_T − NR))/(NI)(5)
where NR_max_T is the maximum theoretical daily nitrogen retention (mg/BWkg^0.67^), which was determined in experiment 1 (3386.61 mg/BW_kg_^0.67^), NR is the daily nitrogen retention (mg/BW_kg_^0.67^) and NI is the daily nitrogen intake (mg/BW_kg_^0.67^).

The concentration of the first limiting amino acid in the dietary protein (*c*) was calculated for each of the diets tested, by the following equation:*c* = 16 × LAAI/NI(6)
where LAAI is the intake of the test amino acid and NI is the nitrogen intake.

The relationship between the efficiency of lysine (reference) and the efficiency of the limiting amino acid under study was used to derive the optimal intake of the amino acid (AAI) (AAI = *bc*^−1^Lys/*bc*^−1^AA), according to Dorigam et al. [14] and Wecke and Liebert [12]. The IAAR was calculated for each amino acid: Met + Cys, Thr, Trp, Arg, Gly + Ser, Val, Ile, Leu, His e Phe + Tyr.

## 3. Results

Assay 1: determination of NMR NR_max_T, and Lys requirement using utilization efficiency.

The results of the nitrogen balance assays are shown in Table 6. The increase in dietary protein levels influenced all the variables studied (*p* < 0.0001). Furthermore, a gradual increase in NI and NEX was observed between the levels of Lys and protein in the diet. However, the ND was gradual up to level five, with a daily difference of only 48.58 mg/BW_kg_^0.67^ between levels five and six, and 35.93 mg/BW_kg_^0.67^ between levels six and seven. The same effect was observed for NMO and NR, with a slight decrease from level five to six (55.22 mg/BW_kg_^0.67^ and 103.80 mg/BW_kg_^0.67^, respectively) and an increase up to level seven (68.12 mg/BW_kg_^0.67^ and 177.94 mg/BW_kg_^0.67^, respectively). The difference in dietary Lys levels provided a range of 78.06% in NMO, which influenced the difference of 68.35% from level one to level seven of Lys in NR.

The exponential function between NI and NEX was used to estimate the NMR (Figure 1). The NMR value obtained was 425.27 mg/BW_kg_^0.67^ per day for quail in the laying period, where NI was equal to zero. With the adjustment of the non-linear regression between NI and NR, it was possible to estimate the value of 3386.61 mg/BW_kg_^0.67^ of daily NR_max_T (Figure 2).

Based on the models used to calculate nitrogen utilization efficiency, that is, the dietary protein quality (*b* = [ln(NR_max_T) − ln(NR_max_T − NR)]/NI), it was possible to estimate the value of *b* = 0.000486. Using the ratio of *b* and the concentration of the limiting amino acid in the protein (*c*), it was possible to estimate the Lys efficiency of utilization Lys *bc*^−1^ = 0.000101. Based on the parameters derived from the models and on the response of Japanese quail breeders in this study, the Lys intake was required to reach 80% of the NR_max_T value, which was 961.90 mg/BW_kg_^0.67^ per day for a bird weighing 0.16 kg Moreover, the daily intake of Lys is 291 mg/bird/day or 1.164% Lys in the diet, considering a feed intake of 25 g/bird.

Assay 2: Determination of amino acid utilization efficiency, requirements, and optimal amino acid ratio.

The results of the nitrogen balance test for Japanese quail breeders are presented in Table 7. From these data, the values of *b* and *bc*^−1^ were derived, which enabled the determination of the IAAR. The birds that consumed diets with amino acid deletion showed, a reduction in nitrogen retention, ranging from 1.47% to 45.69%, in the nitrogen balance, with the addition of NMO. Within this range glycine + serine was the amino acid treatment with the smallest reduction (18.60 mg/BW_kg_^0.67^) and valine was the treatment with the highest (579.98 mg/BW_kg_^0.67^) when compared with CD. These results were influenced by the NI, which differed for the birds that consumed a diet limited in a valine (*p* = 0.0165) and glycine + serine (*p* = 0.0022) compared to that of the CD treatment group. All birds showed an increase in NEX after the treatments with individually limited amino acids, with an overall average that was 31.90% higher than that of the CD treatment group. In particular, birds that consumed a leucine-limited diet had an increased NEX of 42.23% when compared with that of the CD group.

Body weight was compromised in birds that received diets limited in threonine and valine, whereas the others showed no difference compared with that in the CD group (*p* > 0.05). In contrast, the NMO was lower than 376.56 mg/BW_kg_^0.67^ (the mean CD) for all treatments with one limiting amino acid, but only diets limited in lysine and valine differed significantly from CD (*p* < 0.05), with 168.85 and 223.52 mg/BW_kg_^0.67^, respectively.

After a more specific evaluation of the responses using the estimation of parameter b, the limiting diets presented lower protein quality when compared with that of CD (*p* < 0.0001). An exception to this was the diet limited in glycine + serine (*p* = 0.1064). When evaluating the values of *bc*^−1^, the highest use efficiency of an amino acid was for tryptophan (0.000226), and the lowest was for leucine, phenylalanine + tyrosine, arginine, and lysine (Table 8).

The ideal ratio derived from the efficiency of the individual amino acids (*bc*^−1^) in this study, using the Goettingen approach, is presented in Table 8. A 40% reduction was sufficient to estimate the ideal ratio for all essential amino acids tested, except for glycine + serine, owing to the efficiency of use (*bc*^−1^).

## 4. Discussion

To the best of our knowledge, this study is the first to determine the Lys requirement for Japanese quail breeders. We used the nitrogen balance and estimated the NR_max_T, NMR, *b*, and *bc*^−1^ [11,13,14]. Moreover, using dietary protein utilization efficiency, we determined the ideal ratio of essential amino acids based on amino acid deletion.

Birds had lower feed intake amounts at lower concentrations of limiting amino acids (N1 and N2, Table 3), an effect related to the Lys content [18]. Dietary Lys deficiency renders the rate of protein synthesis unfeasible for cell cycle proliferation, leading to cell apoptosis [19]. This can lead to lower ovary and oviduct weight and a consequent decrease in broiler breeders’ production [9]. The reduction in egg production decreases energy requirements, which induces intake regulation [20]. Considering that the energy content at all levels of Lys were the same, the amount of energy required for production would become smaller, and, consequently, the feed intake would be lower. Thus, the lowest concentration of Lys (N1 = 3.4 g/kg and N2 = 5.0 g/kg) made it impossible to maintain production [20]. In this study, a significant increase in egg output was observed up to N5.

NMR is the minimum nitrogen retention, which was estimated considering the intercept of the exponential function when NI = 0 [21]. According to Liebert [21], the NMR value does not consider nitrogen loss from feathers and skin desquamation. Therefore, the NMR results were reported as the approximate average amount of nitrogen [14]. The daily NMR of 425 mg/BW_kg_^0.67^ for Japanese quail breeders was 1.7 times higher than that found for broiler breeders [22]. Using the comparative slaughter technique, Silva et al. [22], found that the NMR is 760 mg/BW_kg_^0.67^ per day for commercial Japanese quails, a value 1.8 times higher than that found in this study for adult birds.

Another important factor causing the difference between the NMR values is the methodology used. NEX determined using the comparative slaughter technique accounts for the nitrogen lost via feathers, which is not included in the nitrogen balance technique [23]. The food restriction imposed in the study conducted by Silva et al. [22], i.e., a lower feed supply (80, 60, and 40%) as the form of limitation, makes homeostasis unfeasible and modifies the anabolic and catabolic responses of the animal. Body proteins are targeted for oxidation and are converted into glucose or ketone bodies for energy generation [24]. Due to the lack of scientific studies determining NMR for Japanese quail breeders, the value found is a reference for other studies (425.27 mg/BW_kg_^0.67^ per day).

The value of 3386.61 mg/BW_kg_^0.67^ is the maximum daily retained nitrogen, under non-limiting conditions, for Japanese quail breeders. This value is expressed as the theoretical limit of the exponential function [13,21]. One application of this constant for nutritional programs is to determine the maximum potential of the strain and allow the estimation of demand according to the production objective. In addition, it is fundamental for practical modelling applications [21].

The estimated NR_max_T values for laying breeders were 1639.9 mg/BW_kg_^0.67^ and 1554.2 mg/BW_kg_^0.67^ for 31–35 and 46–50 weeks, respectively [14], and 1883 mg/BW_kg_^0.67^ for commercial laying hens [15]; both studies used the nitrogen balance methodology. The values approached a 51% difference between the values found for Japanese quail breeders and that for heavy breeders, and the difference decreased slightly to 55% when compared with commercial layers. Comparing NR_max_T results is impossible due to variations in strains, age, feed consumption, and diet characteristics [15,21].

Parameter *b* was estimated as 0.000486, representing the function’s growth rate. The interpretation of b depends on protein quality and is independent of nitrogen intake [21]; the amino acid variation in the protein reduces or increases the amount of nitrogen to its maximum potential [23]. Likewise, calculating the daily requirement of Lys depends only on the efficiency of the dietary amino acid; thus, the parameter *bc*^−1^ was established [25].

The daily value of Lys intake for a bird of 0.16 kg (291 mg/bird) was defined by the efficiency of Lys utilization for the studied diet. This estimated value for the daily intake of Lys reached 80% of the NR_max_T, which was observed in birds of treatment 7 (N7 = 16.8 g/kg of feed). These results characterize the maximum genetic potential of the animal [23,26].

Lys has physiological functions in all cells and tissues during the synthesis of various indispensable compounds [19]. In peak-laying birds, in addition to vital functions, the Lys metabolic pathway, in addition to vital functions, is directed towards yolk and albumen formation, where 87% and 67% of the Lys in the yolk and albumen at peak production, respectively, are from dietary sources [6]. Therefore, Lys intake in the diet is directly related to egg production [9], and insufficient Lys intake makes egg production unfeasible. Establishing Lys requirements is extremely important, as it is considered the reference amino acid to establish the proportions of other essential amino acids [6].

Such proportions were determined from the IAAR based on the deletion of amino acids by the use efficiency of use of dietary protein from the nitrogen balance compared to a CD. Among all the amino acids studied, reduction the of valine in the diet had the biggest influence, reducing feed consumption (30.51%), body weight (12.50%), and nitrogen excreted in the egg output (57.60%), compared with those in birds from the CD group. However, some studies have shown that attention must be paid to the levels of leucine and isoleucine when considering the dietary levels of valine [27,28]. This is because they have a branched chain (BCAA), and their excesses or deficiencies can result in antagonism [29]. Proportional differences were observed in the CD for Leu:Val and Ile:Val at 100:50 and 100:115, respectively, as well in the valine-limited diet where the ratio for Leu:Val and Ile:Val was 100:30 and 100:69, respectively. Therefore, leucine and isoleucine were proportionately higher in the limiting diet. Metabolically, excess leucine induces branched-chain aminotransferase activity, leading to the catabolism of other BCAAs [30]. Valine is the amino acid most susceptible to antagonism and enzymatic degradation [31]. Furthermore, excess leucine stimulates the synthesis of protein and inhibits protein degradation. However, with a deficiency of this amino acid, the stimulation of synthesis further exacerbates the amino acid imbalance in the plasma pool [32]. This corroborates the weight loss observed among the birds, which may be justified by the breakdown of muscle proteins to maintain the plasma balance of amino acids. The detection of an amino acid deficiency in the anterior piriform cortex induces the animal to reduced feed intake [33] as a preventive mechanism. In other studies, the animal reduced the intake of a limiting diet [34,35], which corroborates with the results of this study.

The requirements for the use of an amino acid were divided into the need for maintenance and the efficiency of protein retention, which for birds in production is related to egg output. This was achieved using a factorial approach. This is affected by the deletion of lysine and valine in the diet, via nitrogen deposition, and the logical approach that the 40% limitation made amino acids unavailable for protein synthesis in egg production. In a study by Azzam et al. [36], the non-addition and lower level (1 g/kg) of L-val in the diet significantly affected the serum albumin level. Notably, hepatic production of yolk lipoproteins is regulated by BCAAs [37]. The lysine-related NMO reduction is directly related to egg weight and can be explained by the reduction in egg protein when birds are fed low-lysine diets [38,39]. In lysine metabolism, muscle tissue is manipulated by the rate of egg [6]. In addition, broiler breeders subjected to a dietary lysine deficiency of 44.75% showed a significant difference in egg and chick weights [39]. Kim et al. [9] demonstrated broiler breeder hens fed a 30% lysine reduced diet (0.55% Lys in the diet) had lower oviduct and ovary weights and follicular recruitment. This was accompanied by a delay in ovulation due to apoptosis and necrosis the ovarian follicles, which culminated in a drop in egg production. Therefore, lysine deficiency in the diet makes optimal production impossible because excess lysine is destined for producing eggs only after meeting the muscular requirement [6].

To determine the IAAR, the statistical difference between the individual amino acids and CD must be confirmed in the evaluation of parameter *b* [11]. However, the Gly + Ser-limiting diet did not produce a significant decrease in protein quality when compared to that of CD, indicating an excess of this amino acid in the CD.

The other limiting diets, in terms of individual amino acids, reduced protein quality, which made it possible to estimate the IAA value. Among the studies that recommended amino acid requirements in the literature, only the study by Hanafy and Attia [40] used Japanese quail breeders. However, the recommendation was not estimated (0.2% in the diet) but based on the treatment that resulted in better productivity and reproductive performance. Thus, one can observe the fragility of amino acid nutrition for hens that are fed diets formulated according to recommendations set for commercial laying hens [4,5]. We expect that the dissimilarity in the amino acid requirements is likely due to the difference in genetic potential. When we analysed the recommendation by Rostagno et al. [5], the IAA for all amino acids, except for Val, which was above the recommended value in this study, ranged from 4 (Thr = 0.70%) to 19% (His = 0.48%) among all amino acids studied, except Gly + Ser. According to the NRC [4], the recommended value was transformed into a digestible amino acid considering 89% [16]; therefore, the values for Lys, Met + Cys, Thr, Trp, Arg, Val, Leu, His, and Phe + Try were below those determined in this study (0.79, 0.55, 0.59, 0.15, 0.93, 0.73, 1.12, 0.33, and 1.11% in the diet), with differences reaching 56, 38, and 32% for Met + Cys, Trp, and Val, respectively. Only the IAA for Ile was higher than that for commercial quail (1.12% in the diet). Thus, according to the tables currently used as a basis for formulating rations for breeders, no amino acids presented IAA values close to the results of this study, thereby confirming the modification of the requirement by the genotype.

Furthermore, Lima et al. [41] and Sarcinelli et al. [42] recommended 0.78% and 0.70% of Thr in the diet of Japanese quail, respectively, which corresponds to values 14% and 5% higher than that found in this study (0.67%). Moreover, AAI values were determined in a study by Lima et al. [43], where the estimate for Arg was 17% higher (1.14%) than that found in this study (1.17%). For Trp, they estimated 0.22% in their diet [42], which was 6% higher than the determined value (0.21%). However, the amount of Val in the diet was 62% lower (0.59%) in a study by Martinez et al. [44], in which the authors considered an 11 g/day of egg output and 0.17 kg of body weight. Lower values were also observed for Met + Cys, with a difference of 12% from the estimate by Sarcinelli et al. [42] (0.76% Met + Cys in the diet). No studies have been found in the literature on the IAA for other amino acids.

However, based on the results found in the IAAR of essential amino acids proportional to Lys, breeders need to intake higher proportion of the amino acid intake of Met + Cys, Thr, Trp, Arg, Val, and Ile. For Leu, His, and Phe + Tyr, the recommendations for commercial quails are proportionately higher than those of Rostagno et al. [5]. Thus, when we used the Lys requirement estimated in this study for a 0.16 kg bird (291 mg/bird/day) to determine the IAAR intake of Met + Cys, Thr, Trp, Arg, Val, Ile, Leu, His, and Phe + Tyr as 252, 195, 60, 340, 279, 192, 413, 113, and 387 mg/bird/day, respectively, all amino acids except for His had a higher intake requirement for breeders when compared to that recommended by Rostagno et al. [5] for commercial quails.

Several factors may explain these observations. First, with genetic improvements made in these birds through crosses and specific selections, more efficient commercial birds that need a lower amino acid intake were developed. According to differences related to the methods used to derive the IAAR, studies by Rostagno et al. [5], compiling previous studies, used different diets, ages, and methods to estimate the IAAR with different response criteria. Most of the studies cited in this discussion for IAA are dose-response studies [43,44], unlike the present study, which used a nitrogen utilization model to determine individual efficiency with only an experimental diet.

## 5. Conclusions

Breeders of the Japanese quail have lower dietary efficiency than commercial birds; therefore, the consumption thus far may have been underestimated for these birds. Nevertheless, the current results should not be generalised, and further studies investigating the recommendations for amino acid intake should be conducted to validate these results.

## Figures and Tables

**Figure 1 animals-12-02953-f001:**
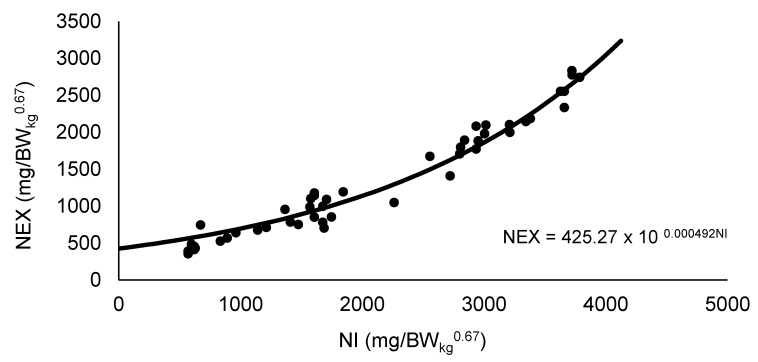
Estimation of the nitrogen requirements for maintenance by fitting an exponential function between the nitrogen intake (NI) and nitrogen excretion (NEX) during a gradual increase in supplied protein limited in lysine for Japanese quail breeders. Values observed (
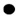
) and predicted (
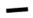
).

**Figure 2 animals-12-02953-f002:**
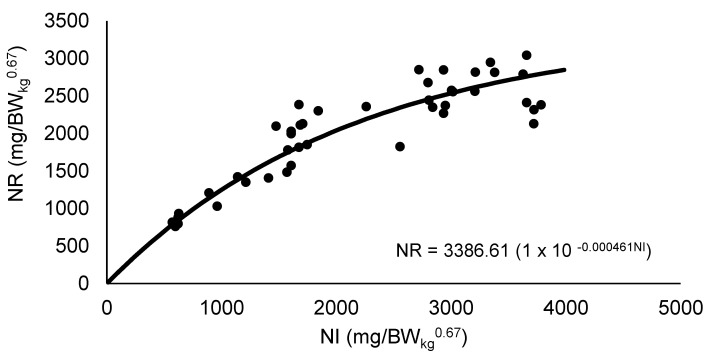
Estimation of the theoretical potential for nitrogen retention in Japanese quail breeders based on the exponential fitting between the daily nitrogen intake (NI) and the daily nitrogen retention (NR). Values observed (
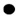
) and predicted (
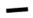
).

**Table 1 animals-12-02953-t001:** Composition (g/kg) of the diets used in the lysine assay.

Ingredient (g/kg)	HPD ^a^	NFD ^b^
Corn	356.97	-
Soybean meal	315.97	-
Corn gluten meal (60% CP)	181.22	-
Soybean oil	20.00	24.84
Dicalcium phosphate	10.13	15.02
Limestone	69.81	69.81
Salt	3.34	3.67
Choline chloride (60%)	0.84	3.40
Mineral premix ^c^	0.25	0.25
Vitamin premix ^c^	0.25	0.25
DL-Met (99%)	4.88	-
L-Lys HCl (78%)	5.72	-
L-Thr	2.71	-
L-Val	3.21	-
L-Ile	2.00	-
L-Arg	10.72	-
LTrp	1.81	-
Potassium chloride	-	11.95
Corn starch	-	249.03
Sugar	-	496.74
Rice husks	-	125.00

^a^ HPD, high protein diet. ^b^ NFD, nitrogen free diet. ^c^ Content per kg of the diet-vit A, 6.668 IU; vit D3, 1.668 IU; vit E, 8 IU; vit K3, 2 mg; vit B1, 1 mg, vit B2, 3.34 mg; vit B6, 2 mg; vit B12, 9 mcg/kg; niacin, 21 mg; chlorine, 0.13 g; pantothenate acid, 8 mg; folic acid, 0.46 mg/kg, biotin, 0.05 mg/kg; 0.46; copper, 8 mg/kg; iron, 6.25 mg/kg; manganese, 70 g; zinc, 25 g; iodine, 6.25 mg; selenium 1.25 mg.

**Table 2 animals-12-02953-t002:** Nutritional levels of experimental diets.

Items	HPD ^a^	NFD ^b^
Calculated composition (g/kg) ^c^		
Metabolizable energy (MJ/kg)	12.5	12.5
Calcium (g/kg)	30.0	30.0
Available phosphorus (g/kg)	3.0	3.0
Analysed composition (g/kg)		
Crude protein	350.0	NI ^e^
Digestible Lys ^d^	16.8	NI
Digestible Met + Cys	17.1	NI
Digestible Met	11.0	NI
Digestible Trp	3.0	NI
Digestible Thr	15.0	NI
Digestible Arg	25.0	NI
Digestible Val	17.0	NI
Digestible Ile	15.0	NI
Digestible Phe	19.0	NI

^a^ HPD, high protein diet. ^b^ NFD, nitrogen free diet. ^c^ The nutrient content of the ingredients used in the formulation was analysed using a near-infrared spectrometer (NIR). ^d^ The total amino acid content of the diets were analysed using HPLC and digestible content calculated using coefficients from Rostagno et al. (2011). ^e^ NI, not identified.

**Table 3 animals-12-02953-t003:** Composition of the control diet (balanced protein).

Items	Content, %
Corn	64.76
Soyabean meal (47%)	12.08
Corn gluten (60%)	5.21
Dicalcium phosphate	1.15
Limestone	7.06
Sodium chloride	0.34
Potassium chloride	0.34
L-lysine (55%)	0.38
DL-methionine (99%)	0.95
L-threonine (98%)	0.27
L-tryptophan	0.10
L-arginine	0.48
L-glycine	0.13
L-valine	0.13
L-histidine	0.15
L-phenylalanine	0.08
L-glutamate	1.00
Choline chloride (60%)	0.16
Premix–Vitaminic ^1^	0.02
Premix–Mineral ^1^	0.02

^1^ Content per kg of the diet-vit A 6.668 IU; vit D3 1.668 IU; vit E 8 IU; vit K 3.2 mg; vit B1 1 mg; vit B2 3.34 mg; vit B6 2 mg; vit B12 5 mcg/kg; niacin 21 mg; chlorine 0.13 g; pantothenate acid 8 mg; folic acid 0.46 mg/kg; biotin 0.05 mg/kg; copper 8 mg/kg; iron 60 g; manganese 70 g; zinc 25 g; iodine 6.25 mg; selenium 0.12 mg.

**Table 4 animals-12-02953-t004:** Composition of the diet for all tested amino acids.

Items	Diets, %										
	Lys	Met + Cys	Thr	Try	Arg	Gly + Ser	Val	Ile	Leu	His	Phe + Try
Balanced diet ^1^	60.00	60.00	59.69	59.86	59.64	59.00	60.00	59.66	59.76	60.00	60.00
Soy oil (47%)	1.15	1.77	1.16	1.15	1.16	1.51	1.15	1.16	1.15	1.15	2.06
Dicalcium phosphate	0.60	0.60	0.60	0.60	0.60	0.62	0.60	0.60	0.60	0.60	0.60
Limestone	2.79	2.80	2.82	2.80	2.82	2.86	2.79	2.82	2.81	2.79	2.79
Sodium chloride	0.15	0.15	0.15	0.15	0.15	0.15	0.15	0.15	0.15	0.15	0.15
Potassium chloride	0.48	0.48	0.48	0.48	0.48	0.49	0.48	0.48	0.48	0.48	0.48
DL-methionine (99%)	0.36	0.00	0.37	0.36	0.37	0.37	0.36	0.37	0.36	0.36	0.36
L-lysine (55%)	0.00	0.80	0.80	0.80	0.80	0.82	0.80	0.80	0.80	0.80	0.80
L-threonine (98%)	0.27	0.27	0.00	0.27	0.27	0.27	0.26	0.27	0.27	0.26	0.26
L-tryptophan	0.09	0.09	0.09	0.00	0.09	0.10	0.09	0.09	0.09	0.09	0.09
L-arginine	0.52	0.52	0.52	0.52	0.00	0.53	0.51	0.52	0.52	0.51	0.51
L-valine	0.34	0.33	0.34	0.33	0.34	0.34	0.00	0.34	0.33	0.33	0.33
L-isoleucine	0.29	0.29	0.29	0.29	0.29	0.30	0.29	0.00	0.29	0.29	0.29
L-leucine	0.67	0.67	0.67	0.67	0.67	0.68	0.67	0.67	0.00	0.67	0.67
L-glycine	0.51	0.51	0.51	0.51	0.52	0.00	0,51	0.51	0.51	0.51	0.51
L-phenylalanine	0.60	0.61	0.61	0.61	0.61	0.62	0.60	0.61	0.61	0.60	0.00
L-histidine	0.19	0.19	0.19	0.19	0.19	0.19	0.19	0.19	0.19	0.00	0.19
L-Glutamate	5.41	4.78	5.40	4.53	6.17	5.51	5.40	4.74	5.18	4.85	4.91
Choline chloride (60%)	0.14	0.14	0.14	0.14	0.14	0.14	0.14	0.14	0.14	0.14	0.14
Maize starch	10.00	5.13	10.00	10.00	10.00	5.44	9.91	10.00	10.00	9.91	4.80
Sugar	6.51	10.00	6.50	6.83	5.82	10.00	6.26	7.07	7.27	6.57	10.00
Inert	8.88	10.00	8.62	8.87	8.82	10.00	8.97	8.76	8.43	8.88	10.00
Premix– ^2^	0.05	0.05	0.05	0.05	0.05	0.05	0.05	0.05	0.05	0.05	0.05

^1^Table 1; ^2^ Content per kg of the diet-vit A 6.668 IU; vit D3 1.668 IU; vit E 8 IU; vit K 3.2 mg; vit B1 1 mg; vit B2 3.34 mg; vit B6 2 mg; vit B12 5 mcg/kg; niacin 21 mg; chlorine 0.13 g; pantothenate acid 8 mg; folic acid 0.46 mg/kg; biotin 0.05 mg/kg; copper 8 mg/kg; iron 60 g; manganese 70 g; zinc 25 g; iodine 6.25 mg; selenium 0.12 mg.

**Table 5 animals-12-02953-t005:** Nutritional levels of experimental diets.

Items	Diets											
	CD	Lys	Met + Cys	Thr	Trp	Arg	Gly + Ser	Val	Ile	Leu	His	Phe + Try
Metabolizable energy (MJ/kg)	11.7	11.7	11.7	11.7	11.7	11.7	11.7	11.7	11.7	11.7	11.7	11.7
Calcium (g/kg)	30.0	30.0	30.0	30.0	30.0	30.0	30.0	30.0	30.0	30.0	30.0	30.0
Available phosphorus (g/kg)	2.8	2.8	2.8	2.8	2.8	2.8	2.8	2.8	2.8	2.8	2.8	2.8
Crude protein (g/kg)	180.1	180.1	180.1	180.1	180.1	180.1	180.1	180.1	180.1	180.1	180.1	180.1
Lysine (g/kg)	10.9	6.6	10.9	10.9	10.9	10.9	10.9	10.9	10.9	10.9	10.9	10.9
Metionine + Cystine (g/kg)	9.0	9.0	5.4	9.0	9.0	9.0	9.0	9.0	9.0	9.0	9.0	9.0
Threonine (g/kg)	6.6	6.6	6.6	3.9	6.6	6.6	6.6	6.6	6.6	6.6	6.6	6.6
Tryptophan (g/kg)	2.3	2.3	2.3	2.3	1.4	2.3	2.3	2.3	2.3	2.3	2.3	2.3
Arginine (g/kg)	12.7	12.7	12.7	12.7	12.7	7.6	12.7	12.7	12.7	12.7	12.7	12.7
Glycine + serine (g/kg)	12.5	12.5	12.5	12.5	12.5	12.5	7.5	12.5	12.5	12.5	12.5	12.5
Valine (g/kg)	8.2	8.2	8.2	8.2	8.2	8.2	8.2	4.9	8.2	8.2	8.2	8.2
Isoleucine (g/kg)	7.1	7.1	7.1	7.1	7.1	7.1	7.1	7.1	4.2	7.1	7.1	7.1
Leucine (g/kg)	16.5	16.5	16.5	16.5	16.5	16.5	16.5	16.5	16.5	9.8	16.5	16.5
Histidine (g/kg)	4.6	4.6	4.6	4.6	4.6	4.6	4.6	4.6	4.6	4.6	2.7	4.6
Phenylalanine + Tyrosine (g/kg)	14.8	14.8	14.8	14.8	14.8	14.8	14.8	14.8	14.8	14.8	14.8	8.9

CD, Control diet.

**Table 6 animals-12-02953-t006:** Mean body weight (BW, kg), feed intake (FI, g/d), daily nitrogen intake (NI, mg/BW_kg_^0.67^), daily nitrogen excretion (NEX, mg/BW_kg_^0.67^), daily nitrogen deposition (ND, mg/BW_kg_^0.67^), nitrogen deposited in egg mass (NMO, mg/BW_kg_^0.67^), and daily nitrogen retention (NR, mg/BW_kg_^0.67^) obtained in nitrogen balance trials with Japanese quail breeders receiving graded levels of protein limitation in lysine.

Diets	N1	N2	N3	N4	N5	N6	N7	DP	*p*-Value
BW (kg)	0.13	0.13	0.15	0.17	0.16	0.17	0.18	0.02	<0.0001
FI (g/d)	15.64	17.76	22.34	22.60	24.36	23.89	23.51	3.60	<0.0001
NI (mg/BW_kg_^0.67^)	611.37	1039.34	1610.75	1610.85	2780.78	3023.17	3645.79	1059.28	<0.0001
NEX (mg/BW_kg_^0.67^)	468.78	625.75	819.56	1094.95	1685.89	1976.86	2563.54	749.89	<0.0001
ND (mg/BW_kg_^0.67^)	178.81	382.25	515.91	791.19	1094.89	1046.31	1082.24	359.69	<0.0001
NMO (mg/BW_kg_^0.67^)	234.29	450.75	740.95	919.71	1055.08	999.86	1067.98	348.92	<0.0001
NR (mg/BW_kg_^0.67^)	838.48	1254.06	1878.61	1957.41	2575.24	2471.44	2649.38	657.35	<0.0001

N1 3.4 g lysine/kg; N2 5.0 g lysine/kg; N3 6.7 g lysine/kg; N3 8.4 g lysine/kg; N5 11.8 g lysine/kg; N6 13.4 g lysine/kg; N7 16.8 g lysine/kg.

**Table 7 animals-12-02953-t007:** Mean values for the effect of exclusion of an amino acid from the diet of Japanese quail breeders in a nitrogen balance trial.

Variables	BW	FI	NI	NEX	NMO	ND	NR	*b* (10^−4^)
Lys	0.15 ^a^	18.62 ^b^	1390.03 ^a^	782.30 ^a^	219.19 ^b^	590.22 ^a^	920.70 ^b^	230 ^b^
Met + Cys	0.15 ^a^	21.14 ^a^	1443.38 ^a^	804.75 ^a^	320.60 ^a^	618.18 ^a^	1054.89 ^a^	258 ^b^
Thr	0.14 ^b^	18.83 ^b^	1412.50 ^a^	866.79 ^b^	249.33 ^a^	572.64 ^a^	934.33 ^b^	225 ^b^
Trp	0.15 ^a^	19.15 ^a^	1443.75 ^a^	840.79 ^b^	239.16 ^a^	602.96 ^a^	965.81 ^b^	236 ^b^
Arg	0.15 ^a^	19.05 ^a^	1411.32 ^a^	853.52 ^b^	332.47 ^a^	557.79 ^b^	1007.84 ^b^	252 ^b^
Gly + Ser	0.15 ^a^	22.43 ^a^	1633.31 ^b^	878.27 ^b^	366.55 ^a^	755.04 ^a^	1241.36 ^a^	280 ^a^
Val	0.14 ^b^	14.78 ^b^	1185.58 ^b^	777.85 ^a^	164.52 ^b^	368.44 ^b^	682.78 ^b^	191 ^b^
Ile	0.14 ^a^	20.18 ^a^	1479.50 ^a^	840.33 ^b^	281.48 ^a^	586.89 ^a^	979.21 ^b^	240 ^b^
Leu	0.15 ^a^	21.39 ^a^	1538.34 ^a^	913.06 ^b^	349.05 ^a^	625.29 ^a^	1094.62 ^a^	254 ^b^
His	0.16 ^a^	22.55 ^a^	1546.02 ^a^	864.84 ^b^	357.82 ^a^	681.19 ^a^	1161.40 ^a^	273 ^b^
Phe + Tyr	0.15 ^a^	19.66 ^a^	1458.94 ^a^	843.05 ^b^	315.82 ^a^	600.69 ^a^	1038.13 ^a^	253 ^b^
DC	0.16 ^a^	21.27 ^a^	1374.43 ^a^	654.44 ^a^	376.56 ^a^	744.63 ^a^	1258.61 ^a^	336 ^a^
Mean	0.15	19.92	1444.00	826.41	301.29	611.84	1040.15	255
SEM	0.001	0.25	15.37	14.10	11.95	15.11	23.17	0.05
*p*-Value	0.0213	<0.0001	<0.0001	0.0195	0.0019	<0.0001	<0.0001	<0.0001

BW, body weight; FI, feed intake (g/day); NI, daily nitrogen intake (mg/BW_kg_^0.67^); NEX, daily nitrogen excretion (mg/BW_kg_^0.67^); NMO, daily nitrogen in egg mass (mg/BW_kg_^0.67^); ND, daily deposited nitrogen (mg/BW_kg_^0.67^); NR, daily nitrogen retention (mg/BW_kg_^0.67^); *b*, slope of the exponential function (protein efficiency); ^a^, ^b^, mean values with ^b^ superscript within the line were significantly different (*p* < 0.05) compared with control diet, by the Dunnett test.

**Table 8 animals-12-02953-t008:** Amino acid utilization efficiency (*bc*^−1^), optimal amino acid intake (AAI) and optimal amino acid ratio (IAAR) for Japanese quail breeders in a nitrogen balance trial.

Variables	*bc* ^−1^	AAI	IAAR
Lys	0.000046	1	100
Met + Cys	0.000056	0.87	87
Thr	0.000074	0.67	67
Trp	0.000226	0.21	21
Arg	0.000042	1.17	117
Gly + Ser	0.000048	-	-
Val	0.000051	0.96	96
Ile	0.000072	0.66	66
Leu	0.000033	1.42	142
His	0.000123	0.39	39
Phe + Tyr	0.000036	1.33	133

## Data Availability

The data can be requested from the corresponding author.

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
