# Peer review of "Determination of the Optimal In-Feed Amino Acid Ratio for Japanese Quail Breeders Based on Utilization Efficiency"

_animals, 2022, doi:10.3390/ani12212953_

Round 1

Reviewer 1 Report

Summary of review

The study aimed to estimate the requirement of essential amino acids for maintenance and the ideal ratio for egg output for quail breeders using the deletion approach. On one hand, the information aimed in this manuscript is very important for the quail industry to improve the offer of nutrients to quails and reduce the waste of nitrogen. On the other, the manuscript is poorly written, demanding a substantial improvement in all topics before publication. The methods used in this study are unusual, which implies a clear and careful description to avoid misinterpretation. The simple summary needs to be rewritten to accommodate the main aspects of this research using simple explanations. The abstract needs further improvements as commented below. The introduction section needs to summarize the problem investigated, but the text needs to be sequential. The introduction was written with various phrases without a connection between them. The material and methods need to be organized. One suggestion is to state at the beginning of this topic that the study was performed with two assays and point out the reason for that. Then, present the method for each assay. It is not necessary to repeat the description of a method. When the authors use the same method, just refer to the text mentioned before. The results and discussion section needs to be reviewed, so the main findings and explanations can be understood by readers. Furthermore, the authors should improve some aspects of the manuscript, which I have specific considerations for.

Specific Reviews

Summary

Line 24: I assume the correct is “feed intake of 25…”

Line 25: It is not clear what the authors want to say with the text “87were determined”. It seems to be the results for the ideal ratio, but the authors should improve the text.

Authors should review the keywords. Perhaps modeling of nitrogen utilization may be changed to two keywords, 1-modeling, and 2- nitrogen utilization.

The last keyword is a whole sentence which is uncommon for keywords. Try to split or use short keywords.

Introduction

Line 38: Lysine is used for egg formation in any laying bird, not only breeders females. Authors should review their text to avoid misunderstanding.

Materials and methods:

Line 69: why did the authors mention only assay 1?

Line 71: How many birds per experimental unit? This information should be cited in the text.

Lines 90-91: In table 2, the amino acids content is given in g/kg. Authors need to be consistent with units. We recommend changing the % for g/kg in the text.

Line 104: change analyzed HPLC to “analyzed using HPLC”.

Line 107: In table 3 the authors give the composition of a control diet. This diet was not presented before, and table 3 was not cited in the text. Why this feed was included in the text?

Line 115: The supply of feed? It is necessary to clarify the text.

Lines 121-122: Is this a total excreta collection technique? Total excreta were collected and weighted. It is not clear only by reading the text.

Line 130: “and nitrogen deposition”

Line 135: It might be better to change to Euler’s number, instead of “basic number of the natural logarithm”

Line 136: It is unnecessary to re-stand the meaning of NEX, NMR, and NI, since it was stated before.

Line 137: The NRmaxT appears at the first time in this line of the text. Thus, authors should include the meaning of these initials.

Line 138: Since the authors will include the meaning for the NRmaxT, it is unnecessary to include it here.

Line 140: The comments about the e made before should be applied here.

Line 140: The meaning for NI was given before, remove it from this line.

Lines 144-147: The same comments are valid for these lines.

Lines 148-152: Please, follow the same logic for the initials in these lines.

Lines 153-154: From the sentence included, it seems the authors used non-linear regression only for data that was not homoscedastic and normal.

I believe that was not what the authors did in their analysis. They need to improve the text to clarify their approach.

Line 158: That is the first appearance of Assay 2, which makes their manuscript quite difficult to follow. Authors should reorganize their material and methods to improve the understanding of their manuscripts.

Line 161: By “refrigerators” the authors want to say air conditioning? Please, reformulate this phrase.

Line 170: “CD” was not stated before in the main text. Please, add the meaning of CD in the first appearance in the text. Note that the abstract is not part of the main text.

Table 5: “DC” is “CD”? Please, standardize throughout your text.

Line 182: Please change “Tables II and III” to “Tables 2 and 3”.

Line 191: include the formula for egg output. “egg output (egg production x egg weight)”.

Line 192: Authors can refer to the methodology used in Assay 1, because they are the same.

Line 203: Which “egg quality variables” were collected?

Lines 210-212: Authors don’t need to include the meaning of the initials here because they are aforementioned.

The equation 7 is not necessary.

Results

Line 232: Change “an amplitude” to “a range”.

Line 242: Change “hens” by “quail”.

Line 247: In figure 1, the Y axes name is misspelled. Change “EXN” to “NEX”.

The X axes name is misspelled. Change “IN” to “NI”.

Line 251: In figure 2, the X axes name is misspelled. Change “IN” to “NI”.

Line 258: Change “the efficiency of use of Lys” to “the Lys efficiency of utilization”.

Lines 161-262: The authors need to improve the text. I assume that 25 g/ bird is the feed intake.

Lines 268-269: Authors state that “a reduction in nitrogen retention ranging from 1.47 to 45.69%”. However, this range demonstrates an increase in nitrogen retention, not a decrease, as stated by the authors.

Table 7: What is “PC”? Is body weight? If yes, it is better to use “BW” instead.

Change “CR” to “FI”.

Change “DC” to “CD”.

Change the comma “,” to dot “.”.

Line 287: them mean 388.04 is not the mean for CD. In table 7 the mean is 376.56.

Furthermore, if lysine and valine were different, why are there no letters to evidence those differences?

Line 295: In table 8, change the comma “,” to dots “.”.

Discussion

Line 321: the word “quail” is repeated in the same sentence.

Line 325: The physiological state of both commercial Japanese quail and Japanese quail breeder is not the same?

Lines 332-334: It is unclear what idea the authors want to give in the text. Perhaps rewriting this sentence might improve the understanding.

Lines 353-362: It is very difficult to follow the authors in this text. Especially at the end of this paragraph. Not sure hose the variation in the population would allow for the intake of Lys…

We suggest the authors rewrite this paragraph to improve text understanding.

Line 367: Please correct the punctuation.

Line 379: Change “DC” to “CD”.

Line 400: The phrase “The egg production rate of determines” does not make sense. Please review the text.

Line 404: Which kind of bird? Quail?

Line 433: Please change “laying hens” to “Japanese quail”.

Line 437: Please remove the comma after the word “it”.

Line 438: Change the word “Martiniz” to “Martinez”.

Author Response

I would like to thank you for all the suggestions made in our manuscript, all changes were followed.

Reviewer 2 Report

 Although the idea of this work is interesting, the authors must reply and respond to these points:

- The number of birds is very low to give a recommendation

- The abstract is not completed. More results should be added and a clear recommendation.

- Any number less than ten should be written in letters (see line 20).

- Revise English thoroughly. 

Author Response

The Journal Animals in it is guideline (https://www.mdpi.com/journal/animals/instructions) calls on the authors to ensure the employment of the 3Rs “…Authors should particularly ensure that their research complies with the commonly accepted '3Rs...”.

Replacement of animals by alternatives wherever possible,

Reduction in number of animals used, and

Refinement of experimental conditions and procedures to minimize the harm to animals.

Several journals define the individual as the smallest experimental unit, among them Poultry Science, Journal Animal Science and others specialized in the area. Therefore, the use of the individual as an experimental unit is supported by the current standards of the Journal Animals.

Additionally, the experimental model used depends on the description of the maximum potential for nitrogen retention - NRmaxT and, it is only possible to determine the maximum with an individual, since groups of individuals generate average values and, in this research, we need the maximum value to characterize the genotype.

Mathematically, the logarithmic transformation assumes that NRmaxT is a greater value than any other value of NR, otherwise the solution of the equation does not result in a real number (set of real numbers - R).

The method developed at the University of Goettingen currently represents the most refined technique to determine the optimal ratio of essential amino acids. A list of publications using this method is presented to demonstrate that in this research we have used a method accepted in the specialised literature.

1) Dorigam JCP, Silva EP, Sakomura NK, Peruzzi NJ, Lima MB, Fernandes JBK. (2020) Alternative procedure for determining lysine maintenance requirement in poultry. Revista Brasileira de Zootecnia 49. Doi: https://doi.org/10.37496/rbz4920180183

2) Soares L, Sakomura NK, Dorigam JCP, Liebert F, Nascimento MQ, Fernandes JBK. (2019) Nitrogen maintenance requirements and potential for nitrogen retention of pullets in growth phase. Journal of Animal Physiology and Animal Nutrition, 103, 1168-1173. Doi: https://doi.org/10.1111/jpn.13117

3) Soares L, Sakomura NK, Dorigam JCP, Liebert F, Sunder A, Nascimento MQ, Leme BB. (2019) Optimal in‐feed amino acid ratio for laying hens based on deletion method. Journal of Animal Physiology and Animal Nutrition 103:1, 170-181. Doi: 10.1111/jpn.13021

4) Lima MB, Sakomura NK, Silva EP, Dorigam JCP, Ferreira NT, Malheiros EB, Fernandes JBK. (2018) The optimal digestible valine, isoleucine and tryptophan intakes of broiler breeder hens for rate of lay. Animal Feed Science and Technology 238, 29-38. Doi: https://doi.org/10.1016/j.anifeedsci.2018.02.001

5) Dorigam JCP, Sakomura, NK, Soares L, Fernandes JBK, Sünder A, Liebert F. (2017) Modelling of lysine requirement in broiler breeder hens based on daily nitrogen retention and efficiency of dietary lysine utilization. Animal Feed Science and Technology 226, 29-38. Doi: https://doi.org/10.1016/j.anifeedsci.2016.12.003

6) Samadi, Wecke C, Pastor A, Liebert F. (2017) Assessing lysine requirement of growing chicken by direct comparison between supplementation technique and “Goettingen Approach”. Open Journal of Animal Sciences 07:01, 56-69. Doi: 10.4236/ojas.2017.71006 

7) Liebert F. (2017) Invited review: Further progress is needed in procedures for the biological evaluation of dietary protein quality in pig and poultry feeds. Archives Animal Breeding 60:3, 259-270. Doi: 10.5194/aab-60-259-2017

8) Wecke C, Pastor A, Liebert F. (2016) Validation of the lysine requirement as reference amino acid for ideal in-feed amino acid ratios in modern fast growing meat-type chickens. Open Journal of Animal Sciences 06:03, 185-194. Doi: 10.4236/ojas.2016.63024 

9) Khan D, Wecke C, Liebert F. (2015) Does the naked neck meat type chicken yield lower methionine requirement data?. Animals 5:2, 151-160. Doi: 10.3390/ani5020151

10) Khan D, Wecke C, Sharifi A, Liebert F. (2015) Evaluating the Age-Dependent Potential for Protein Deposition in Naked Neck Meat Type Chicken. Animals 5:1, 56-70. Doi: 10.3390/ani5010056

11) Khan D, Wecke C, Liebert F. (2015) An elevated dietary cysteine to methionine ratio does not impact on dietary methionine efficiency and the derived optimal methionine to lysine ratio in diets for meat type chicken. Open Journal of Animal Sciences 05:04, 457-466. Doi: 10.4236/ojas.2015.54047

12) Dorigam JCP, Sakomura NK, Hauschild L, Silva EP, Bendezu HCP, Fernandes JBK. (2014) Reevaluation of the digestible lysine requirement for broilers based on genetic potential. Scientia Agricola 71:3, 195-203. Doi: 10.1590/S0103-90162014000300004

13) Silva EP, Sakomura NK, Bonato MA, Donato DCZ, Peruzzi NJ, Fernandes JBK. (2014) Descrição do potencial de retenção de nitrogênio em frangas de postura por diferentes metodologias: mínima retenção. Ciência Rural 44:2, 333-339. Doi: 10.1590/S0103-84782014000200022

14) Pastor C, Wecke F, Liebert. (2013) Assessing the age-dependent optimal dietary branched-chain amino acid ratio in growing chicken by application of a nonlinear modeling procedure. Poultry Science 92:12, 3184-3195. Doi: 10.3382/ps.2013-03340

15) Wecke C, Liebert F. (2013) Improving the reliability of optimal in-feed amino acid ratios based on individual amino acid efficiency data from n balance studies in growing chicken. Animals 3:3, 558-573. Doi: 10.3390/ani3030558

16) Wecke C, Liebert F. (2010) Optimal dietary lysine to threonine ratio in pigs (30-110 kg BW) derived from observed dietary amino acid efficiency. Journal of Animal Physiology and Animal Nutrition 94:6, e277-e285. Doi: 10.1111/j.1439-0396.2009.00969.x

17) Liebert F. (2009) Amino acid requirement studies in Oreochromis niloticus by application of principles of the diet dilution technique. Journal of Animal Physiology and Animal Nutrition 93:6, 787-793. Doi: 10.1111/j.1439-0396.2008.00869.x

18) Kuhi HD, Kebreab E, Lopez S, France J. (2009) Application of the law of diminishing returns to estimate maintenance requirement for amino acids and their efficiency of utilization for accretion in young chicks. The Journal of Agricultural Science 147:04, 383. Doi: 10.1017/S0021859609008442

19) Wecke C, Liebert F. (2009) Lysine requirement studies in modern genotype barrows dependent on age, protein deposition and dietary lysine efficiency. Journal of Animal Physiology and Animal Nutrition 93:3, 295-304. Doi: 10.1111/j.1439-0396.2009.00923.x

20) Liebert F. (2008) Modelling of protein metabolism yields amino acid requirements dependent on dietary amino acid efficiency, Growth Response, Genotype and Age of Growing Chicken. Avian Biology Research 1:3, 101-110. Doi: 10.3184/175815508X388074

- The abstract is not completed. More results should be added and a clear recommendation.

We made these the suggested changes

- Any number less than ten should be written in letters (see line 20).

We made these the suggested changes

- Revise English thoroughly.

We made these the suggested changes. We file the proof.

Reviewer 3 Report

I have suggested a couple of changes (marked in yellow and attached sticky notes)

Author Response

I would like to thank you for all the suggestions made in our manuscript, all changes were followed.

Line 25.

We made these the suggested changes.

Line 99, table 2.

We made these the suggested changes.

Line 107, table 3.

We made these the suggested changes.

Line 130.

We made these the suggested changes.

Line 175, table 4.

We made these the suggested changes.

Line 196, table 8.

We made these the suggested changes.

Round 2

Reviewer 1 Report

Summary of review 2

The authors have done modifications, thank you for the reply. However, this manuscript still needs further changes. In my first review I raised some very important points, that were not fully met. The simple summary still needs improvement. The introduction needs to be sequential. I am not sure why the authors used two separate paragraphs to introduce lysine since their work is about all essential amino acids. This could be improved.

Most of my points were met in material and methods, but I suggest further reading, to improve it even more. I also have specific comments below. The presentation of the methods used, divided into two assays, is much clear now.

Overall, there is some work to be done regarding the English language standards all over the text.

Specific comments:

Line 124: Authors should be more specific about the methodology used to collect the excreta. Did they analyze all excreta collected during the 15 days or a subsample? If it was a subsample, how much? Authors need to have in mind that this work needs to be replicable only by reading the text.

Line 133: “and nitrogen deposition”

Line 137: I suggest removing the word “basic”.

Line 144: the phrase “b is the slope of curve and expresses the quality of the.” suddenly stops; authors should review it.

Line 177: It is Control Diet or Diet Control? Please, review it.

Line: 254: “considering a feed intake of 25g/bird.

Author Response

Summary of review 1

 The authors have done modifications, thank you for the reply. However, this manuscript still needs further changes. In my first review I raised some very important points, that were not fully met. The simple summary still needs improvement. The introduction needs to be sequential. I am not sure why the authors used two separate paragraphs to introduce lysine since their work is about all essential amino acids. This could be improved.

Suggestions were met for the abstract and introduction.

Most of my points were met in material and methods, but I suggest further reading, to improve it even more. I also have specific comments below. The presentation of the methods used, divided into two assays, is much clear now.

Overall, there is some work to be done regarding the English language standards all over the text.

The suggestion was met.

Specific comments:

Line 124: Authors should be more specific about the methodology used to collect the excreta. Did they analyze all excreta collected during the 15 days or a subsample? If it was a subsample, how much? Authors need to have in mind that this work needs to be replicable only by reading the text.

Yes, during the 15-day period, total excreta collection was performed twice a day. For analysis, a sample of approximately 50 grams was taken, identified in the subsequent legend (line 129).

Line 133: “and nitrogen deposition”

We made the correction.

Line 137: I suggest removing the word “basic”.

We made the correction.

Line 144: the phrase “b is the slope of curve and expresses the quality of the.” suddenly stops; authors should review it.

We withdraw this sentence because the meaning of b was explained in the previous paragraph.

Line 177: It is Control Diet or Diet Control? Please, review it.

It is Diet Control. We made the correction.

Line: 254: “considering a feed intake of 25g/bird.

We made the correction.

Reviewer 2 Report

Accept in the current form 

Author Response

Thanks for the suggestions.

The English language has been revised.

Round 3

Reviewer 1 Report

The authors went through some of my suggestions but I still have concerns about the text. Other than that, I deeply suggest an English review of the whole text, done by a specialized company or a native speaker.

Lines 25-26: The way how the authors include the final results is not appropriate (Met+Cys 87, Thr 67, ...). This needs to be better presented to the readers.

Lines 92-93: The composition here is given in % instead of g/kg as in table 2. Authors should standardize the unit.

Line 153: Why authors performed a One-Way Anova? You have a quantitative treatment and the regression analysis was the first goal. Therefore, authors should do only the regression Anova. Because of the dependence between treatments, One-Way Anova is not the best approach. I suggest removing the saying about the One-Way Anova.

Line 350-360: The paragraph is not easy to understand. I have commented before, but the authors did not proceed with further changes.

Line 400: The prase "The egg production rate of determines..." sounds wrong. Please review the text.

Author Response

The authors went through some of my suggestions but I still have concerns about the text. Other than that, I deeply suggest an English review of the whole text, done by a specialized company or a native speaker.

Thanks for the suggestions, they were all followed.
The text was corrected in English, I sent the certificate by e-mail.

Lines 25-26: The way how the authors include the final results is not appropriate (Met+Cys 87, Thr 67, ...). This needs to be better presented to the readers.

We made the changes and kept it as before.

Lines 92-93: The composition here is given in % instead of g/kg as in table 2. Authors should standardize the unit.

We standardized the units, now it is in g/kg.

Line 153: Why authors performed a One-Way Anova? You have a quantitative treatment and the regression analysis was the first goal. Therefore, authors should do only the regression Anova. Because of the dependence between treatments, One-Way Anova is not the best approach. I suggest removing the saying about the One-Way Anova.

Thanks for the suggestion, we removed it.

Line 350-360: The paragraph is not easy to understand. I have commented before, but the authors did not proceed with further changes.

We made changes to make the paragraph easier to understand.

Line 400: The prase "The egg production rate of determines..." sounds wrong. Please review the text.

We revised the text.
